# Immunosuppressive drugs and diet interact to modify the gut microbiota and cardiovascular risk factors, and to trigger diabetes

**Paul Gabarre**[1], **Roberto Palacios**[1], **Kevin Perez**[2], **Philippe Seksik**[3], **Benjamin Bonnard**[1], **Christopher Loens**[1], **Clara Lefranc**[1], **Jean-Paul Pais de Barros**[4], **Louis Anjou**[1], **Yanis Tamzali**[5], **Noël Zahr**[6], **Frédéric Jaisser**[1], **Jérôme Tourret**[7]*

**1** INSERM UMR, Centre de Recherche des Cordeliers CRC, Team Metabolic Diseases, Diabetes and Comorbidities, Paris, France, **2** Department of Biomedical Sciences, University of Lausanne, Lausanne, Switzerland, **3** Gastro-enterology Department, Centre de Recherche Saint Antoine, Sorbonne Université, INSERM UMRS 938, Assistance Publique – Hôpitaux de Paris APHP, Saint-Antoine Hospital, Paris, France, **4** Lipidomic Analytical Platform, INSERM UMR1231 Université de Bourgogne, Dijon, France, **5** Department of Kidney Transplantation – Nephrology, Assistance Publique – Hôpitaux de Paris APHP, Sorbonne Université, Pitié-Salpêtrière Hospital, Paris, France, **6** Department of Pharmacology, Assistance Publique – Hôpitaux de Paris AP-HP, INSERM, CIC-1901, Pharmacokinetics and Therapeutic Drug Monitoring Unit, UMR-S Pitié-Salpêtrière Hospital, Sorbonne Université, Paris, France, **7** Department of Kidney Transplantation – Nephrology, INSERM UMR, Centre de Recherche des Cordeliers CRC, Sorbonne Université, Assistance Publique – Hôpitaux de Paris APHP, Hôpital Pitié-Salpêtrière, Paris, France

* jerome.tourret@aphp.fr

## Abstract

### Background

Kidney transplant recipients are prescribed an immunosuppressive therapy (IST) and some of them follow a high fat diet (HFD) despite medical recommendations. Both are frequently associated with gut microbiota changes and metabolic disorders. We aimed at precisely identifying the effect of the IST and the HFD on metabolic parameters and the gut microbiota in mice, and at establishing correlations between the latters.

### Methods

8-week-old male mice were treated with IST (a combination of prednisone, mycophenolate mofetil and tacrolimus) or not and were fed HFD or standard chow. Metabolic parameters were measured, and the gut microbiota was explored by the quantification of specific bacterial groups by qPCR and by 16S rDNA sequencing.

### Results

The HFD increased insulinemia and decreased the fecal proportion of Bacteroidetes and of *Bacteroides*. The IST increased systolic blood pressure and the fecal proportion of *Escherichia coli*. The HFD and the IST administered together resulted in an additive effect on glucose intolerance, high fasting blood glucose, homeostasis model assessment of

**Data availability statement:** All relevant data are within the manuscript and its Supporting Information files.

**Funding:** This work was funded by grants from the "Institut National de la Santé et de la Recherche Médicale". PG was supported by a grant "Année recherche" from "Assistance Publique – Hôpitaux de Paris AP-HP" in 2018. The funder had no role in study design, data collection and analysis, decision to publish, or preparation of the manuscript.

**Competing interests:** The authors have declared that no competing interests exist.

insulin resistance (HOMA-IR), percentage of fat mass, blood triglyceride, blood cholesterol, and endotoxemia. On the opposite, the HFD and the IST had antagonistic effects on body weight, the proportion of Firmicutes, the Firmicutes/Bacteroidetes ratio, and the proportion of *Clostridium leptum*, *Bifidobacterium*, and *Lactobacillus* in the feces. Finally, we found that the correlations between gut bacterial communities and metabolic consequences of the HFD were altered by the IST.

## Conclusion

The IST and the HFD have specific consequences on the gut microbiota and metabolism. We hypothesize that the metabolic consequences are at least partially mediated by IST/HFD-induced dysbiosis.

## Introduction

Gut dysbiosis is involved in many pathological conditions such as diabetes, metabolic syndrome, inflammatory bowel diseases and cardiovascular diseases [1]. Solid organ transplantation is associated with a persistent intestinal gut dysbiosis defined as an alteration of the gut microbiota (GM) richness, diversity and composition [2]. Many factors are involved including immunosuppressive therapy (IST). Tacrolimus has been shown to induce intestinal dysbiosis [3–6]. Corticosteroids increase the Firmicutes/Bacteroidetes ratio [6]. Finally, Mycophenolate mofetil (MMF) which is the biggest contributor to post transplant digestive disorders, has also been involved in the genesis of a pro-colitogenic gut dysbiosis [7]. We extensively reviewed IST-induced GM dysbiosis [8].

On the other hand, cardio-vascular disease is the leading cause of mortality and graft loss among kidney transplant recipients [9]. It is favored by post-transplant diabetes, hypertension, dyslipidemia [10] which are partly attributable to IST [11,12]. Interestingly, several studies based on murine models have suggested a causative role of IST-induced dysbiosis in the genesis of post-transplant metabolic disorders. For example, in a rat model, Bhat *et al.* corrected tacrolimus induced diabetes and hypercholesterolemia by administering probiotics [5]. We have also shown that *Faecalibacterium prausnitzii* and *Akkermansia muciniphila* concentrations were altered after kidney transplantation (KT) and were associated with the onset of diabetes after KT [13].

The literature about the link between IST-induced GM modifications (or dysbiosis) and post-transplantation metabolic disorders is sparse. Murine models treated with a combination of IST are lacking, and their metabolic consequences are not described. Yet, we have shown that the dysbiosis induced by a combination of IST is different from that induced by each molecules administered separately [6], which suggests that the metabolic repercussions of a combine IS treatment could also be different.

The links between diet, GM dysbiosis and metabolic consequences is abundantly documented [14–16]. Interestingly, not all KT recipients (KTRs) follow diet recommendations after KT [17]. Therefore, it is probable that IST prescribed to all KTRs, and a high fat diet followed by some of them despite medical recommendations, both contribute to GM dysbiosis and metabolic disorders after KT.

The aim of this study was to develop a mouse model treated with a combination of oral immunosuppressive drugs routinely prescribed to KTRs, associated or not with a high-fat diet (HFD), and to precisely describe the impact of each separately or both in combination, on metabolism and the gut microbiota. We also aimed at establishing correlations between metabolic and microbiotic consequences of the IST and the HFD.

## Materials and Methods

### Animal experiments

5-week-old C57BL/6d male mice were ordered from Janvier® laboratory. They were left to acclimatize in the animal facility for 2 weeks before the beginning of any experiment, in order to stabilize their GM.

There were 5 mice per cage. Diets and drug dosages are described below. Mice were divided into four groups as follows. Control mice received standard chow and no immunosuppressive therapy (IST). Mice from the IST group received a combination of tacrolimus, MMF, and prednisolone in the drinking water (see the "diets and treatment administration" paragraph below). Mice from the high fat (HFD) group were given high fat chow. Finally, mice from the IST+HFD group received both the combined IST and HFD.

Metabolic parameters and stools were collected at day 0 (D0) and Day 30 (D30). Stools were immediately frozen until analysis. Because altogether "D0" measures necessitated one week, food and treatments were initiated in 8-week-old mice. More precisely, "D0" data were measured over the 7th week of age. Diets and treatment were then given during the 8th, 9th, 10th, 11th and the 1st two days of the 12th week of age. Altogether this corresponds to 30 days of diet and treatments. "D30" data were collected over the last five days of the 12th week of age. Mice were sacrificed at the beginning of the 13th week of age. Diets and drugs were maintained until sacrifice.

All animal experiments were repeated twice. In all the figure of this manuscript, we present the combined results of the two replicates. In total, there were 15 mice in the control group, 13 mice in the IS+normal chow and IS+HFD groups, and 14 mice in the HFD group without IST.

After 30 days of experimentation, the mice were sacrificed by intraperitoneal injection of ketamine (100 mg/kg) and xylazine (8 mg/kg).

Whole blood was collected by intracardiac puncture and transferred into dry tubes. After centrifugation (10 min at 1000g), the serum was stored at -20 °C.

Animals were kept in a room lighted 12 h per day (7 a.m.–7 p.m.) at a temperature of 22 °C and an average relative humidity of 40%, with *ad libitum* access to chow and filtered tap water. All animal studies were conducted in accordance with the National Institutes of Health Guide and European Community directives for the Care and Use of Laboratory Animals (European Directive, 2010/63/UE) and approved by the local animal ethics committee (agreement #22671-2019103116285394-4). The exclusion criteria were as follows: death, (1-2 mice in each experimental group), excessive weight loss (>20%, no animal excluded), ulcerative dermatitis (no animal excluded). Sample size was determined based on previous studies. In order to avoid sex-biased metabolic interactions, only male mice were used in these experiments.

### Diets and treatment administration

All drug dosages were estimated using food and water consumption charts for C57BL/6d male mice [18]. The standard chow consisted in SAFE A04® pellets, containing 3.1 kcal% fat (SAFE Diets®, France), and HFD consisted in D124912® pellets, containing 60 kcal% fat (Research diets Inc.®, New Brunswick, USA). More precisely, 100g of HFD contained 25.5 g of carbohydrates (of which sugars: 9.5 g), 26 g of proteins, 35 g of fat, 6.5 g of fibers and 6.5 g of minerals. Because tacrolimus (Advagaf®, 5 mg capsules, Astellas pharma®, Japan) is insoluble in water, it was mixed in the chow and pellets were reconstituted. Tacrolimus was given at an estimated dose of 5mg/kg/d in the standard chow groups [19]. Tacrolimus dosage was increased by 50% (30 mg/kg/d) in the HFD groups because of the lower digestive absorption of tacrolimus in case of HFD [20]. MMF (Cellcept® 1g/5 ml powder for oral suspension, Roche®,

Switzerland) and prednisolone (Solupred®, 5 mg pills, Sanofi-Aventis®, France) were given at an estimated dose of 10mg/kg/d [21] and 100mg/kg/d [22] respectively, in the drinking water.

### Clinical data

Mice were weighed once a week. Body fat was measured by MRI using the Bruker® Minispec mq series device (Billerica, Massachusetts, USA) according to the manufacturer instructions.

Systolic blood pressure (SBP) was measured at day 0 and day 30 by photoplethysmography at the tail of the mice with the BP 2000 Blood Pressure Analysis System® (BIOSEB LABinstruments®, Vitrolles, France). Blood pressure was measured 15 times per mouse (5 acclimation measurements followed by 10 consecutive measurements) for 5 consecutive days in the morning. The mean of the last 10 measures of the last 3 days is provided here.

### Biological parameters

Fasting blood glucose and glucose tolerance were measured at day 0 and day 30 during a glucose tolerance test (GTT). After a 6-hour fast, an intraperitoneal injection of glucose (2g/kg) was performed. Blood glucose was measured by tail blood sampling at 0, 15, 30, 60, 90, 120, and 150 min on an Accu-chek Performa® (Roche diabetes France®) blood glucometer. The results were analyzed by comparing the areas under the curve (AUC) of the blood glucose levels of the animals. The higher the AUC, the higher the glucose intolerance (*i.e.,* the lower the glucose tolerance). The HOMA-IR was calculated as follows: insulinemia x fasting blood glucose/ 22.5.

Plasma cholesterol, triglycerides (TG) and blood urea nitrogen were dosed using the Konelab® automatic analyzer, according to the manufacturer's instruction.

Insulinemia was measured by ELISA (Mouse insulin ELISA, Mercodia®) according to the manufacturer's recommendations.

Endotoxemia was estimated by measuring the concentration of circulating lipopolysaccharides (LPS), and precisely the concentration of 3-hydroxymyristate by mass spectrometry as previously described [23]. Mouse plasma (25-50 µL) was mixed with saline to a final volume of 100 µL prior analysis.

Tacrolimus blood samples were analyzed using ultra-performance liquid chromatography (UHPLC) coupled to mass spectrometry (Xevo-TQD®, WATERS® Corp, Milford, MA). Quantifications were achieved in Multiple Reactions Monitoring mode and electrospray ionization was operated in positive mode. Data acquisition was performed using the MassLynx® software. MMF concentrations were measured from plasma by UHPLC method with fluorescence detection (Shimadzu®, Japan). The limit of quantification for Tacrolimus and MMF were 2 ng/mL and 0.2 µg/mL, respectively.

### Gut microbiota analysis

Total DNA was extracted from mouse stools (approximately 50 mg) using the QIAGEN™ Qiamp Powerfecal pro DNA Kit® according to the manufacturer's recommendations with the addition of an initial bead-beating step on a Fast prep 24® (MP biomedical®; 5 cycles of 30 seconds at a frequency of 6 strokes/s). The concentration and quality of the DNA was assessed on a Nanodrop® analyzer and diluted in order to obtain a concentration of 2 ng/µl. Real-time quantitative PCRs were performed with the CFX384 Touch Real-Time PCR Detection System® (BioRad®). Each well contained 5µl of DNA (10 ng), 7.5 µl of 2X SYBR Green, 1.7 µl of water, 0.4 µl of forward primer and 0.4 µl of reverse primer, previously diluted to 10 µM in order to obtain a final concentration of 267 nM each. The sequence of primers used in this study is given in Table 1. All quantifications were done in duplicates.

**Table 1. Sequence of the primers used in this study (5′-3′).**

| Bacteria | Reverse | Forward |
|---|---|---|
| Eubacteria | CGGTGAATACGTTCCCGG | TACGGCTACCTTGTTACGACTT |
| Firmicutes | CAGCAGTAGGGAATCTTC | ACCTACGTATTACCGCGG |
| Bacteroidetes | GCACGGGTGMGTAACRCGTACCCT | GTRTCTCAGTDCCARTGTGGG |
| Bacteroides | CCTWCGATGGATAGGGGTT | CACGCTACTTGGCTGGTTCAG |
| *F. prausnitzii* | CCATGAATTGCCTTCAAAACTGTT | GAGCCTCAGCGTCAGTTGGT |
| *A. muciniphila* | CAGCACGTGAAGGTGGGGAC | CCTTGCGGTTGGCTTCAGAT |
| *E. coli* | CATGCCGCGTGTATGAAGAA | CGGGTAACGTCAATGAGCAAA |
| *Lactobacillus* | AGCAGTAGGGAATCTTCCA | CGCCACTGGTGTTCYTCCATATA |
| *Bifidobacterium* | CGGGTGAGTAATGCGTGACC | TGATAGGACGCGACCCCA |
| *C. leptum* | CCTTCCGTGCCGSAGTTA | GAATTAAACCACATACTCCACTGCTT |
| *C. coccoides* | GACGCCGCGTGAAGGA | AGCCCCAGCCTTTCACATC |

qPCR results are expressed as day 30 to day 0 ratios of fecal proportions of a bacterium X (or bacterial group X) relative to all bacteria in the feces at D30 and D0 respectively. The total amount of bacteria is the result of the qPCR quantification obtained with the "Eubacteria" primers. A D30/D0 ratio < 1 (>1) indicates that the fecal proportion of bacterium X has decreased (increased) between the 30th day of the experiment and the beginning of the experiment. D30/D0 ratios were computed with the following formula:

$$2^{(CtD30Eub - CtD30X) - (CtD0Eub - CtD0X)}$$

Where:

- CtD30Eub (CtD0Eub) is the mean of the two Ct (duplicate qPCR) obtained at D30 (D0) with the Eubacteria primers (quantification of all bacteria)

- CtD30X (CtD0X) is the mean of the two Ct obtained at D30 (D0) with the primers for bacterium X (or bacterial group X)

  The D30/D0 Firmicutes/Bacteroidetes ratio was calculated with the following formula:

$$2^{(CtD30Bac - CtD30Firm) - (CtD0Bac - CtD0Firm)}$$

Where:

- CtD30Bac (CtD0Bac) is the mean of the two Ct obtained at D30 (D0) with the Bacteroidetes primers

- CtD30Firm (CtD0Firm) is the mean of the two Ct obtained at D30 with the Firmicutes primers

## 16S rDNA sequence analysis

The sequences were demultiplexed and quality filtered using the QIIME V.1.9.1 software package. The sequences were then assigned to operational taxonomic units (OTUs) using the UCLUST algorithm with a 97% pairwise identity threshold and classified taxonomically using the Silva reference database (V.132). Rarefaction was performed and used to compare the relative abundance of OTUs across samples. Alpha diversity was estimated using the Shannon and Chao1 diversity indexes. Beta diversity was measured by a Bray-Curtis distance matrix and was used to build principal coordinate analysis plots.

### Statistical analysis

All statistical tests and figures were performed using GraphPad PRISM® (GraphPad Software, 9th version, San Diego, CA 92108, USA) and XLSTAT® (Addinsoft®, Paris, France). The results are presented as mean±standard deviation. Comparisons of clinical and metabolic data between the four groups of mice were tested by one-way ANOVA (normal distribution). The comparisons of D30/D0 ratios of the fecal proportion of bacteria or bacterial groups were tested by Kruskal-Wallis test (skewed distribution) unless otherwise specified. Differences were considered significant if p < 0.05.

## Results

At day 30, tacrolimus and MMF blood concentrations were not significantly different in mice receiving IST with the standard chow or the HFD. The mean drug concentrations were 3.0 ± 1.5 ng/ml and 1.7 ± 1.1 μg/ml in mice receiving the standard chow and 3.3 ± 2.0 ng/ml and 2.1 ± 0.6 μg/ml in mice fed the HFD, respectively (S1 Fig, p = 0.8 and p = 0.5, respectively).

In all the figures, "control" means standard chow without immunosuppressive treatment, HFD stands for "high fat diet", IS for "Immunosuppression".

Hard lines mean p < 0.05 and broken lines mean p < 0.1.

D30/D0 means the ratio of the proportion of a bacterium or a bacterial group after 30 days of treatment and diet to the proportion of this bacterium before the beginning of the treatment and diet.

## Metabolic phenotypes

### Effect of the HFD, elevated insulinemia

Fasting insulinemia was increased in the two groups of mice who received the HFD (HFD and HFD+IST) compared to the control group (Fig. 1). Insulinemia was 68.5 ± 52.2 mUI/L in the HFD group and 104.4 ± 59.4 mUI/L in the HFD+IS group of mice compared to 25.5 ± 10.1 mUI/L in the control group (p = 0.03 and 0.0001, respectively). Insulinemia was also higher in the HFD+IS group than in the IS group (29.3 ± 9.9 mUI/L, 0.0002). Insulinemia tended to be higher in the HFD than in the IS group (p < 0.09).

## Effect of the IS treatment, high systolic blood pressure

The IST was responsible for an increase in SBP after a treatment of 30 days (Fig. 2). The SBP was higher in both group receiving IST (IS group: 142 ± 11 mmHg and HFD+IS group: 145 ± 14 mmHg) than in both groups without IST (controls: 114 ± 10 mmHg and HFD 114 ± 17 mmHg, p < 0.0001 for all comparisons two by two except within treated and non-treated groups). The diet did not impact on SBP.

## SBP: systolic blood pressure

### Additional effects of IST and HFD

**Glucose metabolism.** Fasting blood glucose and insulin resistance (as measured by the HOMA-IR) were not affected by the IST (9.0 ± 6.1 mM and 1.6 ± 1.0, respectively) nor by the HFD (8.0 ± 1.3 mM and 3.2 ± 2.5, respectively) administered separately, compared with controls (6.6 ± 0.9 mM and 1.0 ± 0.3, Fig. 3A and B, p > 0.2 for all comparisons two by two). In contrast, the combination of IST and HFD drastically increased fasting glucose and the HOMA-IR (14 ± 4.0 mM and 8.8 ± 5.3, respectively, p < 0.004 for the comparisons with all the other groups).

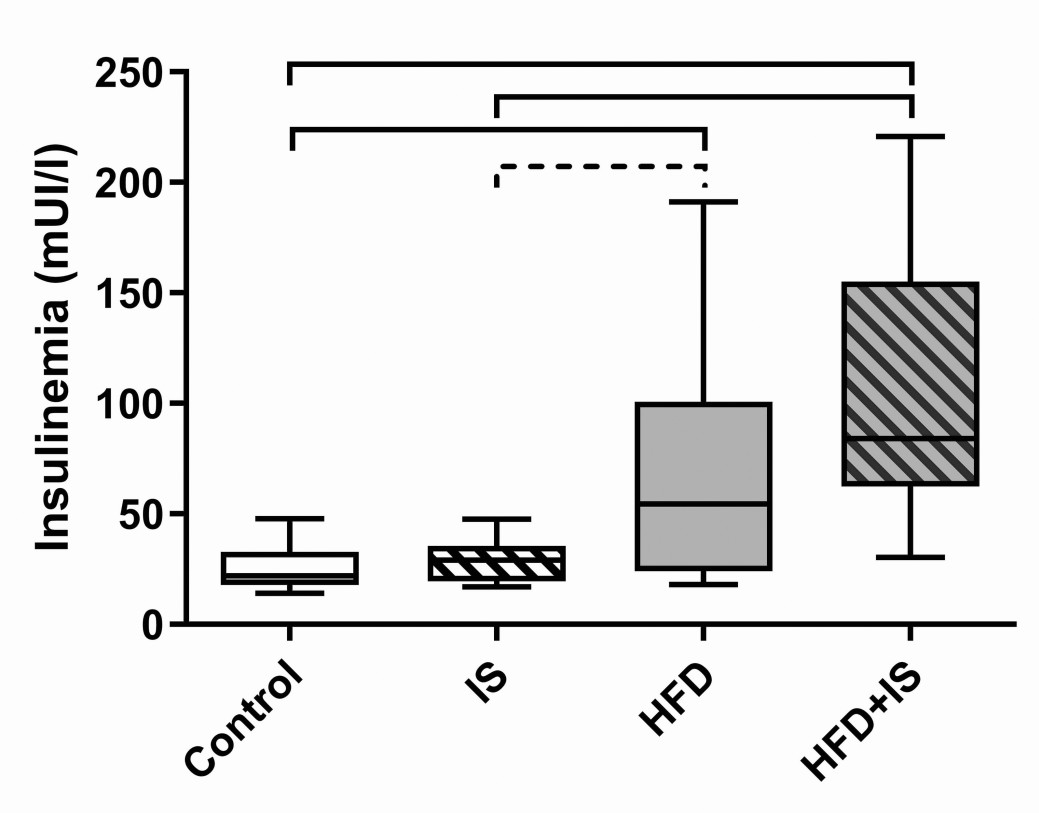

**Fig 1. Effect of the HFD on insulinemia.** The HFD increased insulinemia, whereas the IST had no effect.

Glucose tolerance was assessed by the measurement of glycemia 15 min, 30 min, and then every 30 min for 2h30min after the intraperitoneal injection of 2g/kg of glucose (glucose tolerance test, GTT). The measure was the area under the curve (AUC) of glycemia expressed as a function of time (unit: mmol.l$^{-1}$.min, Fig. 3C). The higher the AUC of the GTT, the higher the glucose intolerance (*i.e.,* the lower the glucose tolerance). The IST was responsible for a glucose intolerance (an increased AUC of the GTT: 1984±896 mmol.l$^{-1}$.min) compared to the control group (931±239 mmol.l$^{-1}$.min, $p<0.0001$) and the group of mice receiving the HFD without IST (1418±378 mmol.l$^{-1}$.min, $p<0.05$). The HFD alone did not significantly affect glucose tolerance. Finally, the combination of HFD and IST was responsible for a higher glucose intolerance than the IST alone (3238±486 mmol.l$^{-1}$.min, $p<0.0001$ for the comparison with the other three groups of mice).

## Lipid metabolism

The percentage of fat mass seemed to be increased by the HFD (11±5.5%) compared to controls (7.7±2.1%) and the mice receiving the IST (7.2±3.5%), but the difference did not reach significance possibly because of an insufficient number of mice ($p=0.1$, Fig. 3D). Only when the HFD was combined with the IST was the percentage of fat mass significantly increased (12.0±3.6%) compared to controls ($p<0.05$) and the IST group ($p=0.03$).

The IST and the HFD did not affect plasma triglyceride concentrations (0.75±0.17 mmol.l$^{-1}$ and 0.65±0.22 mmol.l$^{-1}$, respectively, compared to controls: 0.9±0.2 mmol.l$^{-1}$, Fig. 3E). Only when the IST was combined with the HFD did the triglyceride increase (1.4±0.5 mmol.l$^{-1}$, $p<0.001$ for all the comparison with the other three groups of mice).

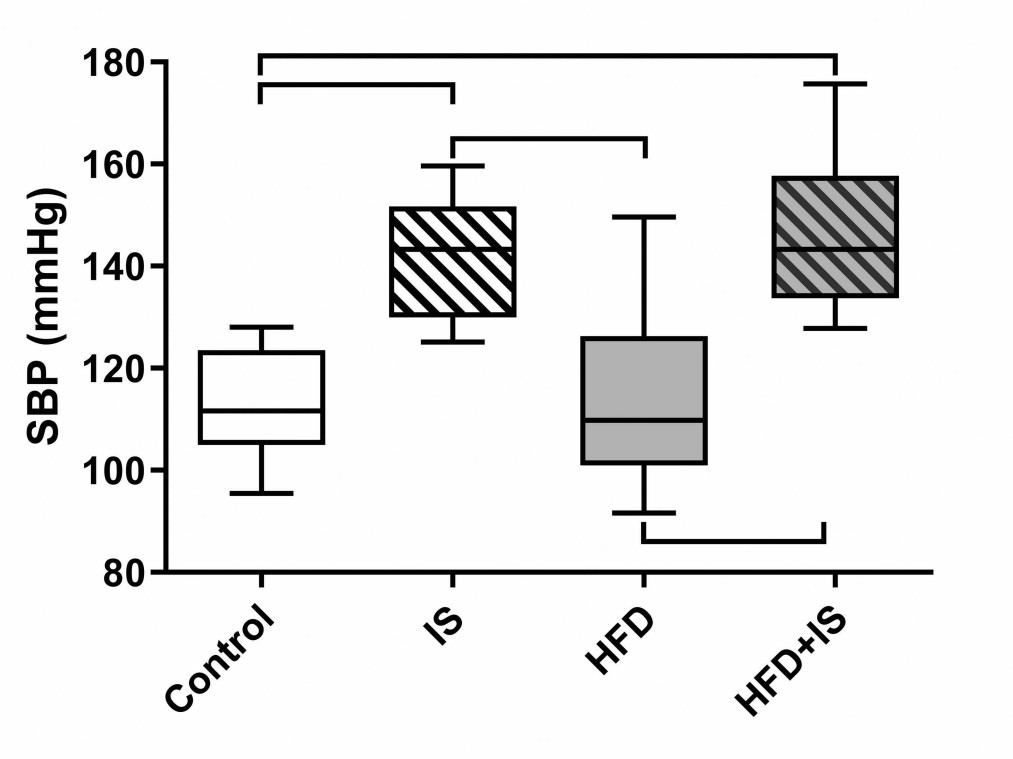

**Fig 2. Effect the IST on systolic blood pressure.** IST increased SBP whereas the diet had no effect.

Finally, both the IST and the HFD increased plasma cholesterol ($3.5 \pm 0.7$ mmol.l$^{-1}$ and $3.3 \pm 0.3$ mmol.l$^{-1}$, respectively) compared to controls ($2.2 \pm 0.3$ mmol.l$^{-1}$, $p < 0.0001$ for both comparison; Fig. 3F). However, when the IST was combined with the HFD, the increase was even more pronounced ($6.0 \pm 1.2$ mmol.l$^{-1}$, $p < 0.0001$ for the comparisons with the other three groups of mice).

## Antagonistic effect of the IST and the HFD

The HFD was responsible for a weight increase ($29 \pm 2.3$ g for mice fed the HFD vs. $26 \pm 1.4$ g for controls, $p = 0.0002$; Fig. 4). On the contrary, the IST was responsible for a weight loss in mice after 30 days of treatment ($20 \pm 1.4$ g for mice receiving the IST, $p < 0.0001$ compared to controls). Finally, mice receiving both the IST and the HFD had an intermediate weight ($22 \pm 1.2$ g): lower than that of controls ($p = 0.0002$) and that of HFD ($p < 0.0001$) but higher than that of mice receiving the IST and normal chow ($p = 0.01$).

## Gut microbiota analysis

### Endotoxemia

The plasma concentration of some components of gut microbial origin – also called "endotoxemia" – has been correlated to micro-inflammation, gut permeability and overall cardiovascular risk [24]. We measured blood levels of lipopolysaccharides (LPS) from the wall of Enterobacteria as a marker of endotoxemia.

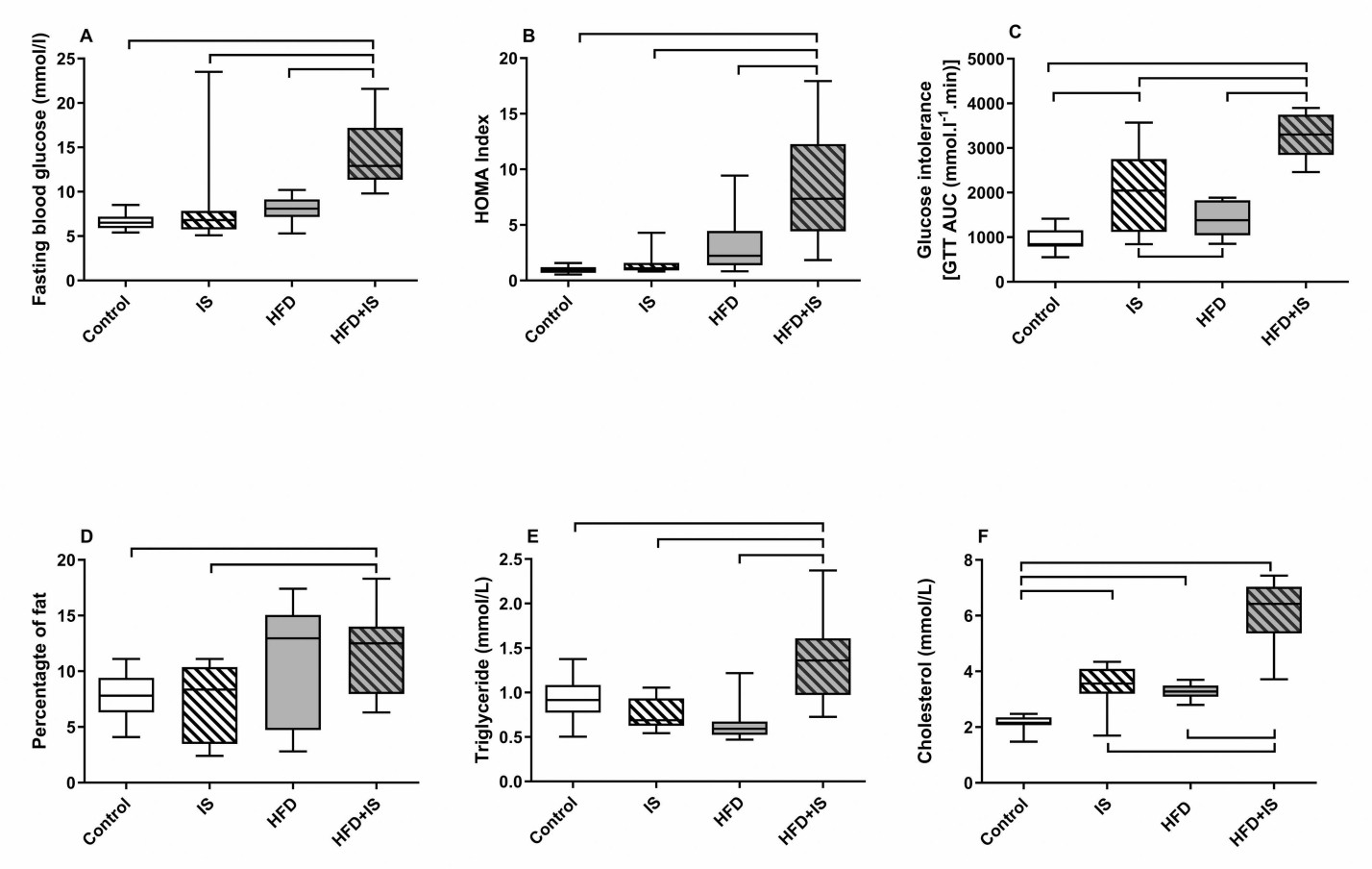

**Fig 3. Additive effects of the HFD and the IST on glucose and lipid metabolism. Glucose metabolism.** Both the HFD and the IST were required to increase fasting blood glucose **(A)** and the HOMA-IR **(B)**. The IST alone (with standard chow) increased glucose intolerance (higher area under the curve (AUC) of the glucose tolerance test (GTT) than in controls, **(C)** but the combination of the IST and the HFD increased it more. **Lipid metabolism.** Both the HFD and the IST were required to increase the percentage of fat **(D)** and blood triglyceride **(E)**. Both the HFD and the IST increased blood cholesterol **(F)** but the combination of the IST and the HFD increased it more.

After 30 days of experiment, endotoxemia as estimated by circulating LPS was increased only when the IST was combined with the HFD (additive effect of the IST and the HFD, Fig. 5). The endotoxemia was $121.2 \pm 38.0$ pmol/ml in the HFD+IS group compared to $56.7 \pm 18.7$ pmol/ml in control mice (p = 0.001) and $63.0 \pm 21.7$ pmol/ml in mice fed with the HFD (p = 0.005). The difference with the group of mice treated with IS ($85.1 \pm 42.3$ pmol/ml) was not significant.

## qPCR quantification of specific bacterial communities

### Effect of HFD

At the phylum level, the HFD decreased the fecal proportion of Bacteroidetes (Fig. 6A). The fecal proportion of Bacteroidetes decreased $2.0 \pm 3.2$ folds between day 30 (D30) and day 0 (D0) in the HDF (which correspond to a D30/D0 ratio of 0.5 on Fig. 6A) and $2.4 \pm 5.0$ folds in the HDF+IS group of mice compared to $0.9 \pm 1.4$ folds in the control group (p = 0.003 and p = 0.0004, respectively).

At the genus level, the proportion of *Bacteroides spp.* decreased (Fig. 7) after 30 days of HFD ($2.4 \pm 5.3$ folds after HFD alone and $2.6 \pm 5.9$ folds after HFD+IS) compared to normal

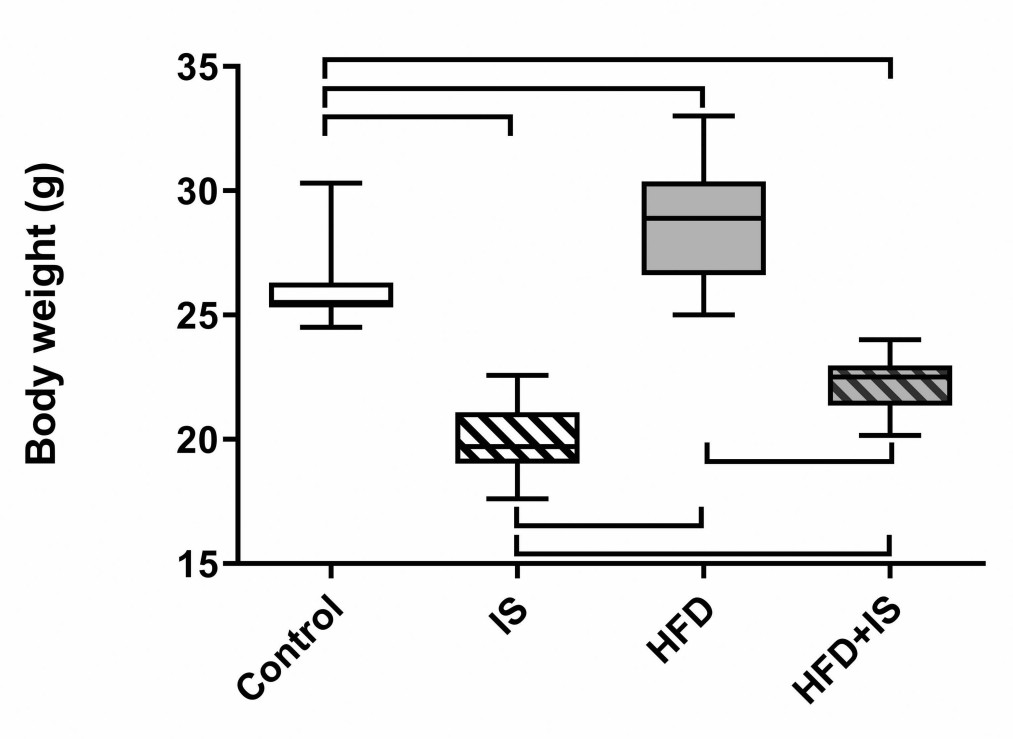

**Fig 4. Antagonistic effect of the HFD and the IST on body weight.** The HFD increased the body weight and the IST decreased it.

chow ($0.9 \pm 1.6$ folds, p = 0.0007 and p = 0.0004, respectively) and normal chow with IST ($0.9 \pm 1.7$ folds, p = 0.002 and p = 0.001, respectively).

## Effect of the IST

The proportion of *Escherichia coli* in the feces was extremely variable from one mouse to the other even in the same cage. This proportion also greatly varied with the diet and the treatment resulting in D30/D0 ratios spanning 6 orders of magnitude. For this reason, *E. coli* D30/D0 ratios were converted in Log10. As the Log10 (*E. coli* D30/D10) had a normal distribution, they were compared with standard one-way ANOVA with turkey's correction for multiple comparisons.

The Log10 (*E. coli* D30/D10) was higher in mice treated with IST ($4.3 \pm 0.8$ in the IST group and $4.0 \pm 1.4$ in the HDF+IST group) than in the feces of mice without IST ($0.6 \pm 0.9$ for controls, p = 0.003 and p = 0.004 respectively; and $1.8 \pm 1.1$ for HFD mice, p = 0.01 for both comparisons; Fig. 8).

## Modulation of the effect of the HFD by the IST

The IST modified the impact of the HFD on most of the bacteria or bacterial groups we chose to quantify.

At the phylum level, the HFD alone increased the proportion of Firmicutes and decreased the proportion of Bacteroidetes, resulting in an important increase of the Firmicutes/Bacteroidetes ratio (Fig. 6A–C). The Firmicutes D30/D0 ratio (Fig. 6B) was $7.0 \pm 8.4$ in the HFD group compared to $2.4 \pm 4.4$ in the control group (p = 0.01) and $1.4 \pm 0.7$ IST group (p < 0.07).

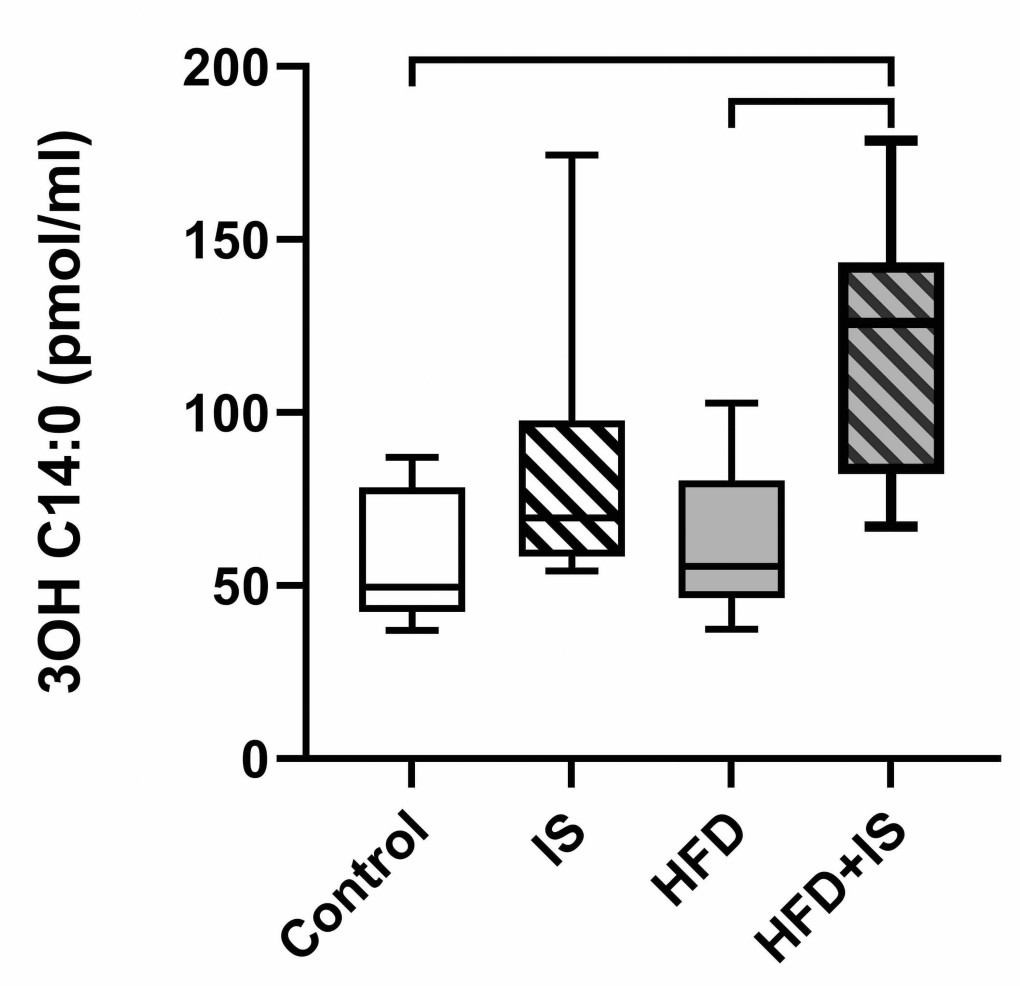

**Fig 5. Additive effects of the HFD and the IST on endotoxemia.** Both the HFD and the IST were required to increase levels of circulating LPS.

In consequence, the Firmicutes/Bacteroidetes ratio (Fig. 6C) increased 24.8 ± 31.1 folds in the HFD group between D30 and D0 (D30/D0 ratio of Firmicutes/Bacteroidetes ratio) compared to 3.6 ± 6.9 folds in the control group (p = 0.002), and 1.9 ± 1.4 in the IS group (p = 0.03). However, the addition of the IST to the HFD counterbalanced the increase in Firmicutes and in the Firmicutes/Bacteroides ratio. This resulted in D30/D0 ratios that were non-significantly different in the HFD+IST mice and in the controls for these two parameters. It also resulted in Firmicutes and Firmicutes/Bacteroidetes D30/D0 ratios that were significantly higher in HFD group than in HDF+IS group (p = 0.0002 and p = 0.09, respectively; Fig. 6B and C).

The *Bidifobacterium spp.* (Fig. 9A) and the *Lactobacillus spp.* (Fig. 9B) D30/D0 ratios were lower in the HFD group than in controls (0.6 ± 0.3 vs. 5.3 ± 6.8, p < 0.0001 and 0.1 ± .11 vs. 1.9 ± 1.6, p < 0.0001, respectively). The IST counterbalanced this effect: the *Bifidobacterium* and the *Lactobacillus* D30/D0 ratios were also lower in the HFD group than in the HFD+IS (2.0 ± 2.2, p < 0.08 and 0.8 ± 0.6, p = 0.04) and the IST normal chow group (1.7 ± 2.3, p < 0.08; and 2.0 ± 1.2, p < 0.0001, respectively).

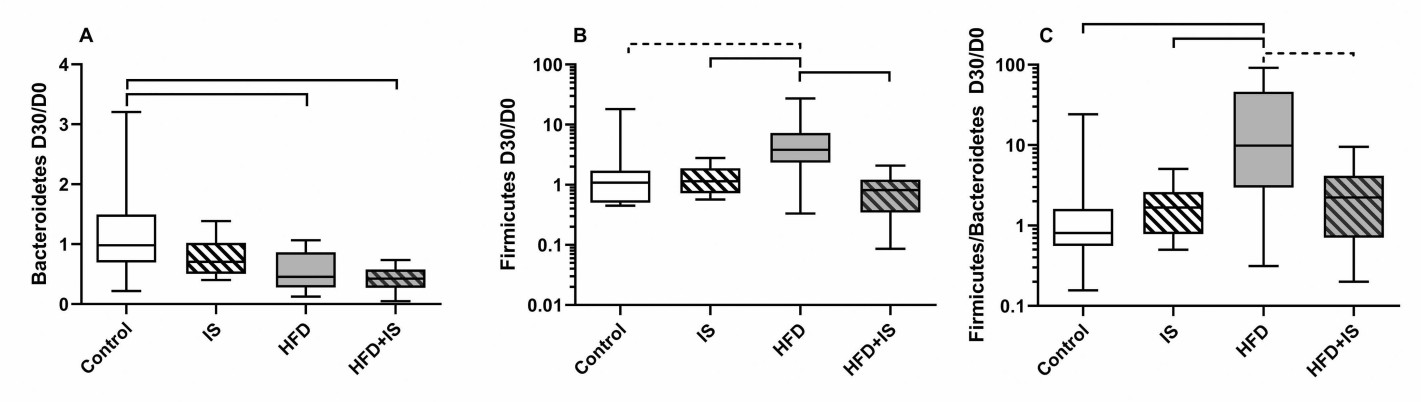

**Fig 6. Antagonistic effects of the HFD and the IST on the gut microbiota at the phylum level.** The HFD decreased the fecal proportion of Bacteroidetes (quantified by qPCR, **A**) and increased the proportion of Firmicutes (**B**) resulting in an increased Firmicutes/Bacteroidetes ratio (**C**). The IST counterbalanced the impact of the HFD on Firmicutes proportion and Firmicutes/Bacteroidetes ratio.

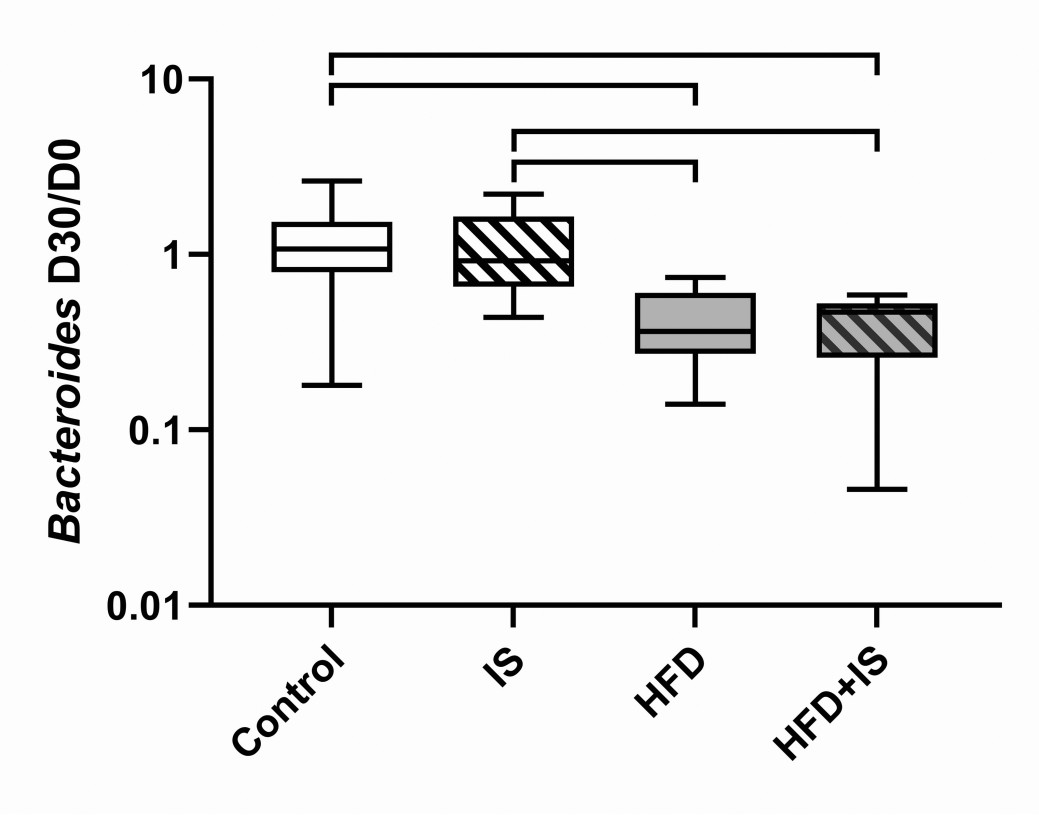

**Fig 7. Effect of the HFD on *Bacteroides*.** The HFD decreased the fecal proportion of *Bacteroides*. The IST had no impact.

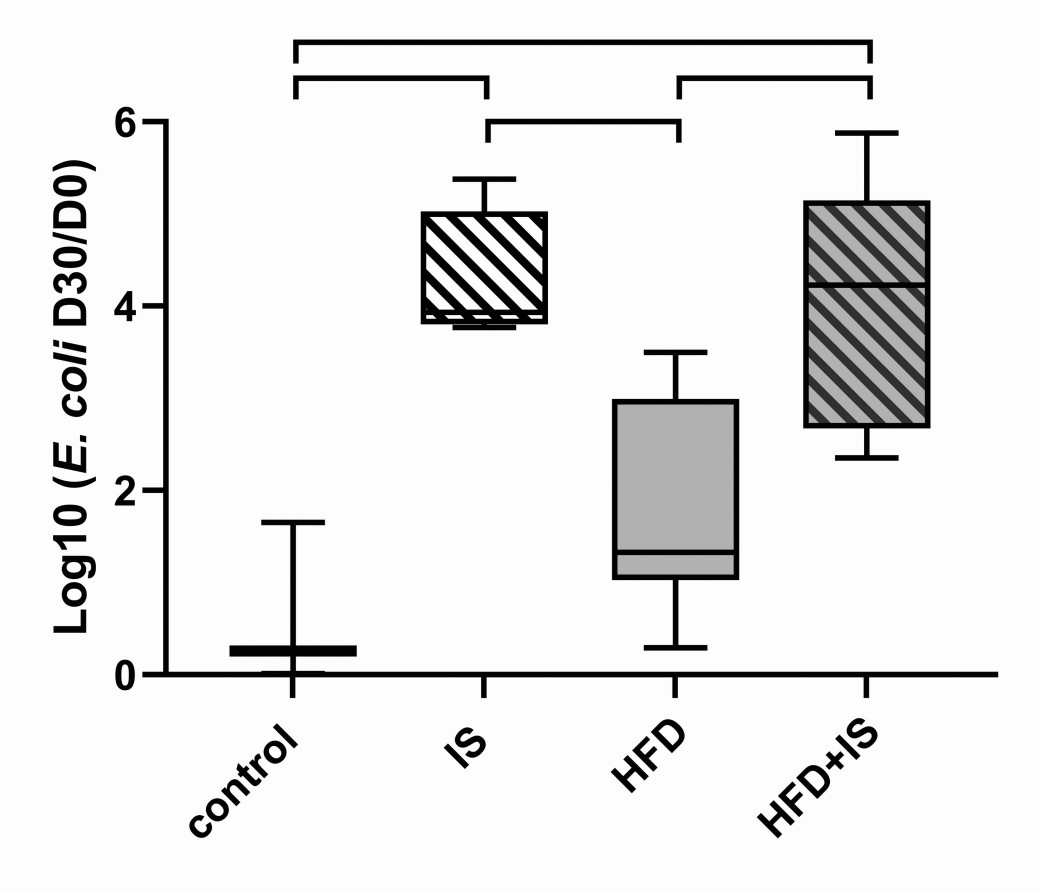

**Fig 8. Effect of the IST on *Escherichia coli*.** The IST increased the fecal proportion of *E. coli*.

For *Clostridium leptum* (Fig. 9C), the profile was the opposite: the *C. leptum* D30/D0 ratio was higher in the HFD group (3.3 ± 2.0) than in all the other groups (1.2 ± 0.9 for controls, p = 0.03; 0.9 ± .03 in the IST group p < 0.006; and 3.1 ± 5.5 in the HFD+IST group, p < 0.1)

The IST and the HFD seemed to have opposite effects on *Clostridium coccoides* (Fig. 8D) and *Lachnospiraceae spp.* (Fig. 9E)

## 16S rDNA sequencing

The gut microbiota was further explored by 16S rDNA deep sequencing of bacterial DNA extracted from the feces.

The alpha diversity measured by the Shannon index was higher in the group of mice receiving both the HFD and the IST than in the mice receiving the HFD only (p < 0.002) and in controls (p < 0.001, Fig. 10A). The alpha diversity found in the feces of the mice that received the IST alone (normal chow) was not significantly different than that in the other groups with great differences between individuals.

Differences in microbial community composition between groups were investigated using weighted UniFrac distance. The resulting distance matrix was visualized by nonmetric multidimensional scaling plots (Fig. 10B). The overall microbial community was significantly different between mice receiving the HFD (with or without IST) and mice receiving standard chow (with or without IST). The two diets were correctly segregated on the horizontal axis

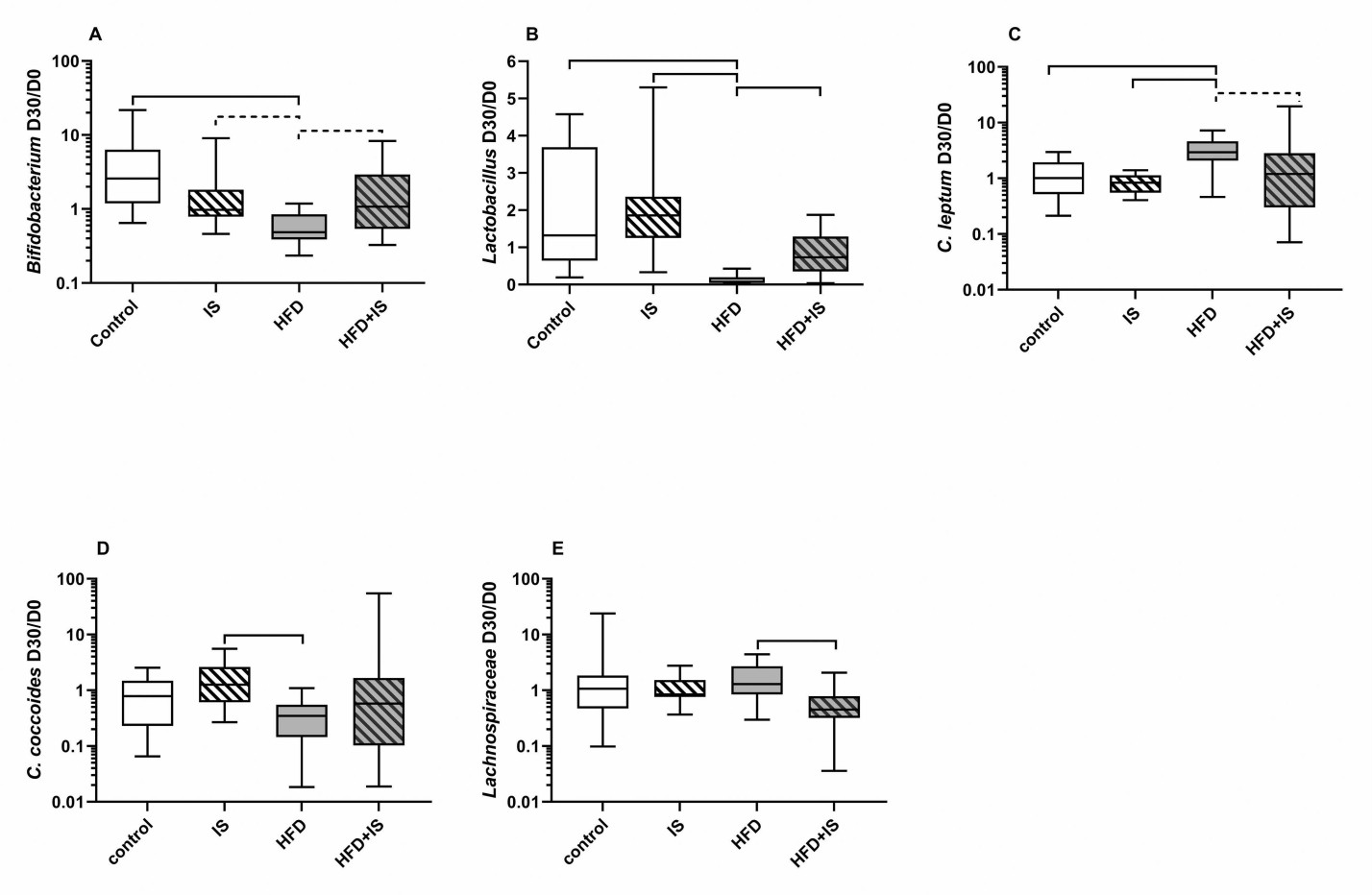

**Fig 9. Modulation by the IST of the impact of HFD on specific bacteria (or bacterial groups) quantified by qPCR.** The HFD decreased the fecal proportion of *Bifidobacterium* (**A**) and *Lactobacillus* (**B**) and increased the proportion of *Clostridium leptum* (**C**). The HFD and the IST seemed to have opposite effects on *Clostridium coccoides* (**D**) and *Lachnospiraceae* (**E**).

(PC1, Fig. 10B). The overall microbial community was not significantly different between mice receiving the standard chow with or without IST. On the opposite, the global composition of the gut microbiota of the mice receiving the HFD and the HFD+IS was different, as shown by a segregation on the vertical axis (PC2). This is another indication that if the IST alone has milder impact on the gut microbiota than the major effect of the HFD, it modulates the diet-induced microbial modifications. The same segregations were observed on the principal component analysis designed with the results of the qPCR quantifications (S2 Fig).

The Firmicutes/Bacteroidetes ratio was significantly increased in mice receiving the HFD ($p < 0.05$). The IST had no significant impact on this increase (Fig. 10A).

At the class level (Fig. 10D), the difference in gut microbiota composition was due to a higher proportion of Clostridia and a lower proportion of Bacteroidia in mice fed the HFD than in mice fed standard chow. Among mice receiving the HDF, the mice receiving the IST showed more Gamma- and fewer Deltaproteobacteria than mice receiving the HFD without IST.

## Correlations between phenotypes and the gut microbiota

We then used the quantifications of fecal bacteria or bacterial groups by qPCR to establish correlations between metabolic parameters and gut bacteria abundance in mice receiving the IST or not.

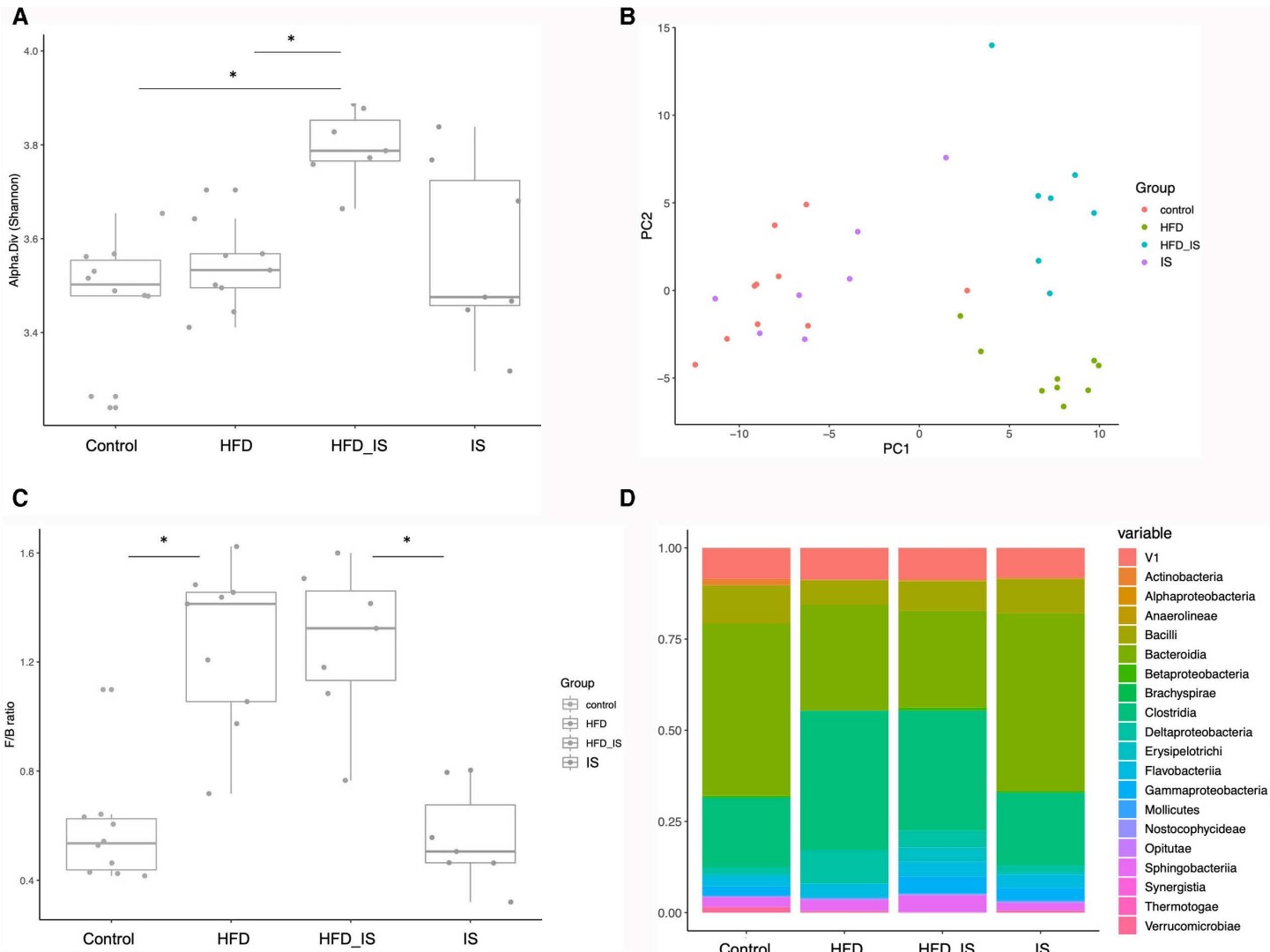

**Fig 10. Gut microbiota analysis by 16S rDNA sequencing.** The association of the HFD and IST increased alpha diversity **(A)**. The F1 axis of the principal coordinate analysis **(B)** discriminated HFD vs. standard chow. The IST modified the HFD-induced dysbiosis as seen by a segregation of the HFD and the HFD+IST groups on the F2 axis. The HFD increased the Firmicutes/Bacteroidetes ratio **(C)**. **D:** Impact of the HFD and the IST on the gut microbiota at the family level.

Figure 11A represents a heat map of correlations between metabolic phenotypes and the relative abundance of specific bacterial communities in the group of mice that did not receive IST (comparison of controls with the HFD group). The fecal proportion of Firmicutes and *C. leptum*, and the F/B ratio were positively correlated with body weight, insulin resistance (HOMA-IR), and fat mass. Similarly, the proportion of *C.coccoides* was positively correlated with triglyceridemia. On the opposite, the proportion of Bacteroidetes, *Bacteroides*, *Lactobacillus*, and *Bifidobacterium* were negatively correlated with these metabolic parameters. Interestingly, the LPS level was positively correlated with glucidic parameters such as high fasting blood glucose and glucose intolerance.

The colors range from red (negative correlation) to green (positive correlation). The asterisks indicate which correlations are significant.

The second heat map (Fig. 11B) represents the same correlations performed in the groups of mice which received the IST (comparisons between the IST and the HFD+IST groups).

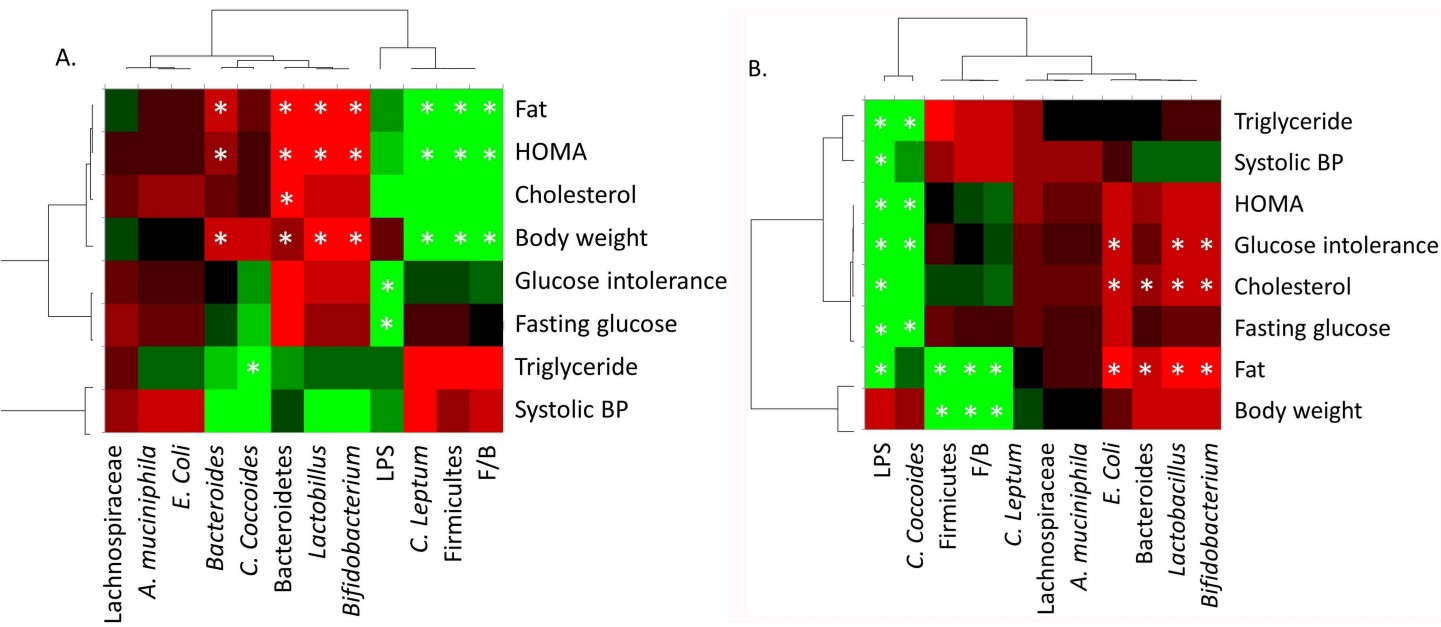

**Fig 11. Heat maps representing the correlations between phenotypic and microbiotic data induced by the HDF without (A) or with (B) IST.** The IST modified the correlations induced by the HFD.

The proportion of Firmicutes, *C. leptum*, and F/B ratio were no longer correlated with insulin resistance. The positive correlation between glucidic parameters and LPS was maintained but in addition there was a positive correlation with high systolic blood pressure, hypercholesterolemia and hypertriglyceridemia. The proportion of *Bacteroides*, *Lactobacillus* and *Bifidobacterium* maintained their negative correlation with fat mass but were no longer negatively correlated with body weight and insulin resistance. In addition, they were negatively correlated with cholesterolemia.

Because these two heat maps cannot be overlaid, we conclude that it is possible that the addition of an IST to the HFD alters the interactions between the gut microbiota and metabolism.

## Discussion

We developed a model of mice treated with an association of oral immunosuppressive drugs in association or not with a high-fat diet. We chose this model because we believe it parallels the real life experience of KTRs. All of them receive a combined IST (frequently consisting of an association of prednisone, MMF and/or tacrolimus in most of the cases [25] and some of them follow a high calorie diet despite dietetic recommendations. As both have consequences on the GM and cardiovascular risk factors, our model aimed at untangling in mice, the specific effects of the combined IST and diet on the GM and on metabolism.

To our knowledge, this is the first study to precisely detail the metabolic consequences of the IST, the HFD or both and to correlate these modifications to the gut microbiota. We found that the IST and the HFD profoundly modify metabolic parameters as well as the gut microbiota. We were able to evidence the specific effects of the IST and of the HFD. More interestingly, we were able to evidence the interactions between the IST and the HFD, which could be neutral, positive, or negative. All the metabolic and gut microbiota modifications found in this study are summarized in the Table 2.

**Table 2. Summary of the effects of the IST, the HFD and the combination of both compared to controls.**

| Parameter | IS | HFD | HFD+IS | Interaction* IS/HFD | Figure |
|---|---|---|---|---|---|
| Insulinemia | ↔ | ↗ | ↗ | None | 1 |
| Bacteroidetes, *Bacteroides* | ↔ | ↘ | ↘ | None | 6A, 7 |
| Systolic blood pressure, *E. coli* | ↗ | ↔ | ↗ | None | 2, 8 |
| Glucose intolerance | ↗ | ↔ | ↗ | Additive | 3C |
| Fasting blood glucose, HOMA, % fat, TG, endotoxemia | ↔ | ↔ | ↗ | Additive | 3A, 3B, 3D, 3E,5 |
| Cholesterol | ↗ | ↗ | ↗ | Additive | 3F |
| Body weight | ↘ | ↗ | ↘ | Antagonistic | 4 |
| Firmicutes, Firmicutes/Bacteroidetes, *Clostridium leptum* | ↔ | ↗ | ↔ | Antagonistic | 6B, 6C, 9C |
| *Bifidobacterium, Lactobacillus* | ↔ | ↘ | ↔ | Antagonistic | 9A, 9B |

*: an interaction (additive or antagonistic) is defined by the fact that the combination of HDF and IST significantly modified the considered parameters compared to both the HFD and the IST administered separately.

As expected, mice receiving the HFD developed metabolic complications such as hyper-cholesterolemia and insulin resistance [26]. More interestingly, the IST worsened these abnormalities. In some cases, the metabolic effect was observed only in mice receiving both the IST and the HFD.

The combination of HFD and IST resulted in increased levels of LPS. Flanningan *et al.* also found that serum LPS was augmented in mice treated with MMF [7]. In this study and in our previous work [6], we found that the IST was responsible for a proliferation of gamma-Proteobacteria, and especially *E. coli*, which produces LPS. The increased endotoxemia probably is the result of this proliferation and could also be the consequence of an increased gut permeability. Endotoxemia is known to promote metabolic disorders by enhancing systemic inflammation [27]. Elevation of LPS levels is also associated with a diminution of graft survival [28] and cardiovascular events after KT [27].

The combination of the IST and the HFD was also required to observe diabetes and hypertriglyceridemia. IS drugs may precipitate the onset of metabolic disorders induced by the HFD: tacrolimus has a direct toxicity on the pancreatic Beta cells and decreases insulin secretion [29]. Glucocorticoids increase hepatic insulin resistance, visceral fat deposit and lipolysis [30]. However, we hypothesize that IS-induced gut microbiota alterations could also be involved in the genesis of these metabolic abnormalities. Indeed, we observed that the HFD was associated with a gut dysbiosis that has already been described [31]. At the phylum level, we observed an increase in the proportion of Firmicutes, a decrease in the proportion of Bacteroidetes, and consequently an increase in the F/B ratio between day 0 and day 30. At the species level, we found a reduction in the proportion of *Bacteroides*, *Lactobacillus* and *Bifidobacterium* in the HFD group. These bacteria are involved in the regulation of insulin resistance, body fat, circulating LPS and short chain fatty acid levels [32–34]. The IST alone slightly altered the microbiota compared to the impact of the HFD; only *E. coli* seemed to be increased by the IST alone, as we previously reported [6]. In contrast, the IST seemed to significantly alter the interaction between the HFG and the gut microbiota. For example, the IST prevented the increase in the proportion of Firmicutes associated with the HFD and thus normalized the F/B ratio. The IST also annulled the modifications induced by the HFD on *C. leptum*, *Bifidobacterium*, and *Lactobacillus*. It is possible that these effects have GM-induced metabolic consequences.

As a consequence, we found that IST altered the correlations between the gut microbiota and metabolic modifications induced by the HFD. We believe that this study is clinically

relevant because it sheds a new light to the pathophysiology of IS-induced metabolic disorders. Indeed, after KT some patients develop diabetes and other do not. It is possible that new onset diabetes after transplantation requires a specific association of three cofactors: a specific microbiota [13], an IST and a HFD. It is possible that a predisposing microbiota ("predysbiotic microbiota" [13]) resulting from a HFD would be modified by IST to reach the dysbiosis known to trigger diabetes. The participation of the gut microbiota in metabolic complications after KT could explain the great variability of the phenotypes observed in humans as these could depend on the pretransplant gut microbiota of the patients.

The weight loss in mice treated with IS drugs was unexpected. It may be related to the effect of MMF which induces colitis and malabsorption [7]. However, mice receiving MMF did not present diarrhea. Stools which were regularly collected from all the mice had a normal texture in IST mice as well as in mice without IST. Mice receiving IST also did not seem to be dehydrated as they presented high blood pressure (Fig. 2). Even though we used drug dosages that have already been used in the literature, this weight loss could be an indication of IST toxicity. At sacrifice at the end of the first replicate, we performed whole blood and plasma dosage of tacrolimus and MMF respectively. Mean whole blood of tacrolimus was not significantly different in the IS and the HFD+IS groups. The mean tacrolimus concentration for all treated mice was $3.2 \pm 1.7$ µg.L$^{-1}$ (S1 Fig). Considering that drugs were continuously delivered through beverage and food, this leads to an estimated area under the curve over 24h (AUC$_{0-24}$) of 3.2 x 24 = 77 µg.L$^{-1}$.h (not taking into account that mice eat more during the night and that sacrifices were performed in the morning). This is far below what is considered as therapeutic in humans [35]. Similarly, the mean MMF plasma concentration of all IST treated mice was $1.9 \pm 0.9$ µg.L$^{-1}$ leading to an estimated area under the curve over 12 hours of 1.9 x 24 = 45.6 µg.L$^{-1}$.h. This is within therapeutic level in humans [36]. Therefore, we found no indication of IST toxicity in our experiments.

Our model has advantages. First, it describes a robust model to study the impact and the interaction of diet and the IST. The use of the oral route for treatment spears the mice the stress of oral gavage, therefore reducing bias on stress-depending parameters such as fasting blood glucose, blood pressure and the gut microbiota [37]. The HFD modifies the intestinal absorption of tacrolimus, but this was circumvented in our study by an increase of tacrolimus concentration in food pellets. MMF and tacrolimus blood dosage performed at sacrifice showed non-significantly different concentrations of both molecules in IS drug treated mice with HDF or standard chow (S1 Fig). The metabolic phenotypes that we observed are described in detail and are robust, since we obtained the same results in two replicates of the same experiment. Furthermore, there are several reports on the effect of IST on the gut microbiota (that we reviewed in [8]) but very few using a combined IST paralleling what is done in routine transplantation clinics.

Our study also has limitations. The model we use does not take into account many interfering factors associated with kidney transplantation. Among those, we can cite factors associated with alloimmunity. The presence of an organ with a different HLA genotype bears consequences on the gut microbiota that we could not assess[38]. However, rodent models of kidney transplantation remain difficult to implement and introducing a 3$^{rd}$ variable (alloimmunity in addition to diet and IST) would probably have blurred the conclusions. Because kidney failure can alter the gut microbiota [39] and because some mice were treated with the nephrotoxic drug tacrolimus, we measured blood urea nitrogen in all mice before treatments and diets were started and before sacrifice, at day 30. Blood urea nitrogen was not significantly different between mice groups and timepoints (S3 Fig). We decided to focus on the conjointly deleterious effect of HFD and IST; in the opposite, we did not explore if a high fiber diet could protect from the deleterious effects

of IST. Also, KTRs receive numerous anti-infectious agents (cotrimoxazole, valgancyclovir, antibiotic cures for infections) which have an important impact on the GM. It would be extremely complex to develop an animal model which comprises all the parameters associated with KT. This is why we decided to isolate the impact of two predominant factors (diet and IST) that have a strong impact on the GM in order to study their interaction. The four-week duration of diets and treatments may seem short, especially when studying lipid metabolism. However, most of metabolic modifications occur very shortly (sometimes immediately) after KT. Notwithstanding the short duration of the experimentation, we were able to evidence clear-cut modifications induced by HFD, IST or both. Therefore, we believe that the present *in vivo* study is very consistent with what we observe in KTRs. The delivery mode of the IST was continuous as opposed to the BID prescription of MMF and tacrolimus and the QD prescription for prednisone in patients. It is possible that this continuous delivery has a different impact on metabolism and the gut microbiota than discontinuous delivery. Finally, this work only establishes correlations between phenotypic and microbiotic abnormalities without exploring the possible causal link. This is a field of exploration *per se* as the links between the IST-induced microbiota modifications and their possible consequences are extremely wide as they involve metabolomic (short-chain fatty acids, free fatty acids, bile acids, hydrogen sulfide), immunology and molecular biology of the cell. Fecal microbiota transfer experiments with the feces of mice receiving or not the IST and/or the HFD are underway to explore if the microbiota reproduces the metabolic consequences. This is the first step to go from the present correlative study to causative evidence.

## Conclusion

We have shown that the IST and HFD can mimic both phenotypic and gut microbiota abnormalities commonly observed in kidney transplant recipients. This raises the question of the role of the gut microbiota in the genesis of metabolic disorders induced by IS treatments and diet.

## Supporting information

**S1 Fig: Whole blood tacrolimus and MMF plasma concentration in IST mice according to the diet.** Even though HFD decreases tacrolimus absorption, our dose adaptation resulted in comparable drug concentration.
(JPG)

**S2 Fig: Principal component analysis designed with the the qPCR quantifications of specific bacteria in the gut microbiota (see Table 1 for the bacterial species or groups that were quantified by qPCR).** The overall composition of the gut microbiota as assessed by qPCR quantification of specific bacteria was significantly different in the high fat diet group compared to the standard diet group, with or without immunosuppressive drugs (segregation on the F1 axis). However, the adjunction of immunosuppressive drugs to the high fat diet modulated the overall microbiota composition. Indeed, the HFD and HFD+IS groups were not segregated by the F1 axis but by the F2 axis. These results were similar to those obtained with the 16S rDNA sequencing of gut microbiota.
(JPG)

**S3 Fig: Kidney function estimated by BUN.** BUN was not significantly different throughout the experiment and between groups. BUN: Blood urea nitrogen.
(JPG)

## Acknowledgement

We thank the team of the CEF of the Cordeliers Research Center for their support with animal care.

We thank Alexander Bartsev for his logistic help and coordination of the works performed at Lebiome®.

## Author contributions

**Conceptualization:** Paul Gabarre, Roberto Palacios, Philippe Seksik, Christopher Loens, Clara Lefranc, Jean-Paul Pais de Barros, Yanis Tamzali, Noël Zahr, Frédéric Jaisser, Jérôme Tourret.

**Data curation:** Paul Gabarre, Roberto Palacios, Kevin Perez, Benjamin Bonnard, Christopher Loens, Clara Lefranc, Jean-Paul Pais de Barros, Louis Anjou, Noël Zahr, Jérôme Tourret.

**Formal analysis:** Paul Gabarre, Roberto Palacios, Kevin Perez, Philippe Seksik, Benjamin Bonnard, Clara Lefranc, Jean-Paul Pais de Barros, Noël Zahr, Frédéric Jaisser, Jérôme Tourret.

**Funding acquisition:** Paul Gabarre, Noël Zahr, Frédéric Jaisser, Jérôme Tourret.

**Investigation:** Roberto Palacios, Benjamin Bonnard, Christopher Loens, Clara Lefranc, Jean-Paul Pais de Barros, Louis Anjou, Yanis Tamzali, Noël Zahr, Jérôme Tourret.

**Methodology:** Paul Gabarre, Roberto Palacios, Philippe Seksik, Benjamin Bonnard, Clara Lefranc, Jean-Paul Pais de Barros, Yanis Tamzali, Noël Zahr, Frédéric Jaisser, Jérôme Tourret.

**Project administration:** Frédéric Jaisser, Jérôme Tourret.

**Software:** Kevin Perez.

**Supervision:** Roberto Palacios, Philippe Seksik, Frédéric Jaisser, Jérôme Tourret.

**Validation:** Jérôme Tourret.

**Writing – original draft:** Paul Gabarre, Jérôme Tourret.

**Writing – review & editing:** Paul Gabarre, Roberto Palacios, Kevin Perez, Philippe Seksik, Benjamin Bonnard, Christopher Loens, Clara Lefranc, Jean-Paul Pais de Barros, Louis Anjou, Yanis Tamzali, Noël Zahr, Frédéric Jaisser.

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
