## [Decision Letter · Decision Letter 0]

17 Jul 2024

PONE-D-24-11620Immunosuppressive drugs and diet interact to modify the gut microbiota and cardiovascular risk factors, and to trigger diabetesPLOS ONE

Dear Dr. Tourret,

Thank you for submitting your manuscript to PLOS ONE. After careful consideration, we feel that it has merit but does not fully meet PLOS ONE’s publication criteria as it currently stands. Therefore, we invite you to submit a revised version of the manuscript that addresses the points raised during the review process. As raised by one reviewer, your observatioins on the effect of HFD is not very clear. You need at least to rerun some measurments on plasma TG and Cholesterol levels. It seems that your HFD is only for 4 weeks which is quite short in order to increase plasmatic TG and CHOL levels. You have to clearly justify the duration of the diet. In this situation, you cannot speak about  combine effects of HFD and IS.

We look forward to receiving your revised manuscript.

Kind regards,

Catherine Mounier

Academic Editor

PLOS ONE

“This work was funded by grants from the “Institut National de la Santé et de la Recherche Médicale”.

PG was supported by a grant “Année recherche” from “Assistance Publique – Hôpitaux de Paris AP-HP” in 2018.”

Reviewers' comments:

Reviewer's Responses to Questions

**Comments to the Author**

1. Is the manuscript technically sound, and do the data support the conclusions?

Reviewer #1: No

Reviewer #2: Yes

2. Has the statistical analysis been performed appropriately and rigorously?

Reviewer #1: No

Reviewer #2: Yes

3. Have the authors made all data underlying the findings in their manuscript fully available?

Reviewer #1: Yes

Reviewer #2: Yes

4. Is the manuscript presented in an intelligible fashion and written in standard English?

Reviewer #1: No

Reviewer #2: Yes

5. Review Comments to the Author

Reviewer #1: General comment

The values for total cholesterol and triglycerides presented in the manuscript are not appropriate for C56Bl6 mice (please see, DOI:10.1016/j.yexcr.2015.07.032) therefore you should re-run all the measurements and the statistical analyses, as well. Additionally, the data reported about the loss of BW indicate that the drug treatment was toxic for the mice. The Authors have included a sentence in this regards in the Discussion, but I believe that is not sufficient because they did not take into account this possibility.

Reviewer #2: Kidney transplantation requires a very thorough medical care post-surgery, including a specific diet and immunosuppressive therapy. Solid organ transplantation may lead to a change of the microbiota, with long-term consequences. This study is relevant both to the clinician in their daily practice, but also as a steppingstone for further studies, emphasizing the importance of all factors that may impact the metabolism of patients with kidney transplants. The acquirement of diabetes mellitus in such patients may have negative impact on quality of life, increasing morbidity and mortality.

The study is well conducted, clearly elaborated and explained in the manuscript, with appropriate statistical variables and analysis. The conclusion of the study is well supported, while the limitations have been clearly stated. All the necessary data and scientific findings have been made available in the manuscript. The text presents a high level of written clarity, with concise and clear ideas that provide the manuscript an optimal level of comprehension given the complex subject. The manuscript has been very well researched and referenced. The use of tables and figures are useful in increasing the understanding of the study. Particularly, Table 2 offers a concise summary of some of the aspects of the study and facilitates the readability of the manuscript.

After analyzing this manuscript, it can be considered for publication

6. PLOS authors have the option to publish the peer review history of their article (what does this mean? ). If published, this will include your full peer review and any attached files.

**Do you want your identity to be public for this peer review?** For information about this choice, including consent withdrawal, please see our Privacy Policy .

Reviewer #1: No

Reviewer #2: **Yes: ** Elena Rezus

---

## [Author Response · Author response to Decision Letter 1]

28 Oct 2024

Editor

As raised by one reviewer, your observatioins on the effect of HFD is not very clear. You need at least to rerun some measurments on plasma TG and Cholesterol levels.

≫ The lipid values are in international units, mmol/L and not in mg/dL as stated on the graph. We apologize for this mistake but there is nothing wrong with the dosages that we performed. The graph now shows mmol/L.

It seems that your HFD is only for 4 weeks which is quite short in order to increase plasmatic TG and CHOL levels. You have to clearly justify the duration of the diet. In this situation, you cannot speak about combine effects of HFD and IS.

≫ Even though 4 weeks seems a short duration for the diet and the IST, we were able to observed clear-cut differences between groups. We chose this duration because new onset diabetes, weight gain, HBP, dyslipidemia are observed rapidly after KT as a consequence of IST. Also, we wanted to distinguish direct effects of diet and IST from long terms effect that maybe more related to weight gain and insulin resistance. We added a comment in the discussion.

Reviewer #1: General comment

The values for total cholesterol and triglycerides presented in the manuscript are not appropriate for C56Bl6 mice (please see, DOI:10.1016/j.yexcr.2015.07.032) therefore you should re-run all the measurements and the statistical analyses, as well.

≫ The lipid values are in international units, mmol/L and not in mg/dL as stated on the graph. We apologize for this mistake but there is nothing wrong with the dosages that we performed. The graph now shows mmol/L.

Additionally, the data reported about the loss of BW indicate that the drug treatment was toxic for the mice. The Authors have included a sentence in this regards in the Discussion, but I believe that is not sufficient because they did not take into account this possibility.

≫ We have added IS drug blood dosages in the discussion. Based on these dosages, we explain why we don’t think that IST were toxic in our experiments.

---

## [Decision Letter · Decision Letter 1]

7 Jan 2025

PONE-D-24-11620R1Immunosuppressive drugs and diet interact to modify the gut microbiota and cardiovascular risk factors, and to trigger diabetesPLOS ONE

Dear Dr. Tourret,

Thank you for submitting your manuscript to PLOS ONE. After careful consideration, we feel that it has merit but does not fully meet PLOS ONE’s publication criteria as it currently stands. Therefore, we invite you to submit a revised version of the manuscript that addresses the points raised during the review process.

Some comments raised by a reviewer needs to be adressed before final acceptance

We look forward to receiving your revised manuscript.

Kind regards,

Catherine Mounier

Academic Editor

PLOS ONE

Journal Requirements:

Reviewers' comments:

Reviewer's Responses to Questions

**Comments to the Author**

1. If the authors have adequately addressed your comments raised in a previous round of review and you feel that this manuscript is now acceptable for publication, you may indicate that here to bypass the “Comments to the Author” section, enter your conflict of interest statement in the “Confidential to Editor” section, and submit your "Accept" recommendation.

Reviewer #2: All comments have been addressed

Reviewer #3: (No Response)

2. Is the manuscript technically sound, and do the data support the conclusions?

Reviewer #2: Yes

Reviewer #3: Partly

3. Has the statistical analysis been performed appropriately and rigorously?

Reviewer #2: Yes

Reviewer #3: No

4. Have the authors made all data underlying the findings in their manuscript fully available?

Reviewer #2: Yes

Reviewer #3: Yes

5. Is the manuscript presented in an intelligible fashion and written in standard English?

Reviewer #2: Yes

Reviewer #3: Yes

6. Review Comments to the Author

Reviewer #2: Immunosuppressive treatment may play an important role in determining metabolic disfunction. Analyzing the combination of administered immunosuppressive therapeutic agents and high fat diet may provide information regarding the real-life experience of kidney transplant recipients. The subject of the study is relevant in clinical practice and original. The study presents research results that have not been published elsewhere. The methodology of the manuscript is well elaborated and explained, with a high level of the provided information and sufficient details, including statistical analysis. The information from this manuscript renders the process replicable. The conclusions are succinct, but present the main points of the study and are supported by the supplied data. The manuscript is logically structured, with an appropriate level of written clarity. The language is clear and concise, and uses suitable terminology given the context. Overall, the written comprehension of the manuscript is optimal. The research elaborated in the manuscript respects ethical standards regarding research on animal models and there are no elements that suggest transgression of research integrity. The results have been appropriately reported as per the required standards. The data concerning the study have been made fully available for analysis.

After analyzing this manuscript, it can be considered for publication.

Reviewer #3: This study showed that immunosuppressive therapy (IST) and high-fat diet (HFD) had some effects on gut microbiota and metabolism in mice. Here are some comments.

1. It could be improved by mentioning the specific immunosuppressive drugs used in the Abstract.

2. Note the units of insulin concentration , which should be consistent in the main text and Figure 1; The ordinate of Figure 5 is consistent with the main text.

3. LPS on line 247 is not equivalent to endotoxemia.

4. Specify the exact IST regimen used (prednisone, MMF, tacrolimus, etc.) and their doses.

5. Provide more details on the HFD composition.

6. Clarify the duration of the experiment and the time points for data collection.

7. Include more statistical details in figure legends.

8. A more in-depth discussion of the mechanisms by which IST and HFD may alter the gut microbiota and impact metabolism is necessary.

9. A discussion of the limitations of the study and potential areas for future research is necessary.

7. PLOS authors have the option to publish the peer review history of their article (what does this mean? ). If published, this will include your full peer review and any attached files.

**Do you want your identity to be public for this peer review?** For information about this choice, including consent withdrawal, please see our Privacy Policy .

Reviewer #2: **Yes: ** Elena Rezus

Reviewer #3: **Yes: ** Shijun Yue

---

## [Author Response · Author response to Decision Letter 2]

13 Feb 2025

Rebuttal letter

Reviewer #2:

After analyzing this manuscript, it can be considered for publication.

≫ Thank you very much

Reviewer #3: This study showed that immunosuppressive therapy (IST) and high-fat diet (HFD) had some effects on gut microbiota and metabolism in mice. Here are some comments.

1. It could be improved by mentioning the specific immunosuppressive drugs used in the Abstract.

≫ This precision was added to the revised manuscript

2. Note the units of insulin concentration, which should be consistent in the main text and Figure 1; The ordinate of Figure 5 is consistent with the main text.

≫ We apologize for this mistake and sincerely thank reviewer #3 for noticing it. The unit for insulin concentration is now consistent in the text and the figure: mUI/L.

3. LPS on line 247 is not equivalent to endotoxemia

≫ This was specified in the main text

4. Specify the exact IST regimen used (prednisone, MMF, tacrolimus, etc.) and their doses.

≫ We believe that this is already detailed in the paragraph “diets and treatment administration” of the material and methods section

5. Provide more details on the HFD composition

≫ This was added to the manuscript

6. Clarify the duration of the experiment and the time points for data collection

≫ This has been detailed in the material and methods section of the present revision

7. Include more statistical details in figure legends.

≫ The statistical details are described in the material and methods section and mostly comprises simple comparison tests. Adding them in the legend of each figure depends on the editorial choices of each journal and this was not asked to us in the first round of revisions.

8. A more in-depth discussion of the mechanisms by which IST and HFD may alter the gut microbiota and impact metabolism is necessary.

≫ “The mechanisms by which IST and HFD may alter the gut microbiota and impact metabolism” is a whole field of research per se and is way beyond the scope of the research presented here. Consequently, we believe that describing these mechanisms more in details would only be speculative as we did not investigate them, and wouldn’t add any scientific value to our work. However, we have already cited several works that specifically address these questions in the introduction and the discussion (see references 6-8, 13, and 31-34).

9. A discussion of the limitations of the study and potential areas for future research is necessary.

≫ This is the second round of revisions. A discussion of the limits of our work was already included in the original submission. It was extended in the first revision of the manuscript. Please tell us if you believe that a specific limitation has not been addressed in the last paragraph of the discussion.

As for future research, we specify in the manuscript that microbiota transfer experiments are underway. We believe that the goal of a scientific paper is more to describe what has been done than what is intended to be done.

---

## [Decision Letter · Decision Letter 2]

19 Feb 2025

Immunosuppressive drugs and diet interact to modify the gut microbiota and cardiovascular risk factors, and to trigger diabetes

PONE-D-24-11620R2

Dear Dr. Tourret

We’re pleased to inform you that your manuscript has been judged scientifically suitable for publication and will be formally accepted for publication once it meets all outstanding technical requirements.

Kind regards,

Catherine Mounier

Academic Editor

PLOS ONE

Additional Editor Comments (optional):

Reviewers' comments:

Reviewer's Responses to Questions

**Comments to the Author**

1. If the authors have adequately addressed your comments raised in a previous round of review and you feel that this manuscript is now acceptable for publication, you may indicate that here to bypass the “Comments to the Author” section, enter your conflict of interest statement in the “Confidential to Editor” section, and submit your "Accept" recommendation.

Reviewer #3: All comments have been addressed

2. Is the manuscript technically sound, and do the data support the conclusions?

Reviewer #3: Yes

3. Has the statistical analysis been performed appropriately and rigorously?

Reviewer #3: Yes

4. Have the authors made all data underlying the findings in their manuscript fully available?

Reviewer #3: Yes

5. Is the manuscript presented in an intelligible fashion and written in standard English?

Reviewer #3: Yes

6. Review Comments to the Author

Reviewer #3: (No Response)

7. PLOS authors have the option to publish the peer review history of their article (what does this mean? ). If published, this will include your full peer review and any attached files.

**Do you want your identity to be public for this peer review?** For information about this choice, including consent withdrawal, please see our Privacy Policy .

Reviewer #3: **Yes: ** Shi-Jun Yue

---

## [Editor Report · Acceptance letter]

PONE-D-24-11620R2

PLOS ONE

Dear Dr. Tourret,

I'm pleased to inform you that your manuscript has been deemed suitable for publication in PLOS ONE. Congratulations! Your manuscript is now being handed over to our production team.

Kind regards,

on behalf of

Dr. Catherine Mounier

Academic Editor

PLOS ONE